# LEARNING REDUCED FLUID DYNAMICS

## ABSTRACT

Predicting the state evolution of ultra high-dimensional, time-reversible fluid dynamic systems is a crucial but computationally expensive task. Model reduction has been proven an effective method to reduce computational costs by learning a low-dimensional state embedding. However, existing reduced models are irrespective of either the time reversible property or the nonlinear dynamics, leading to sub-optimal performance. We propose a model-based approach to identify locally optimal, model-reduced, time reversible, nonlinear fluid dynamic systems. Our main idea is to use stochastic Riemann optimization to obtain a high-quality reduced fluid model by minimizing the expected trajectory-wise model reduction error over a given distribution of initial conditions. To this end, our method formulates the reduced fluid dynamics as an invertible state transfer function parameterized by the reduced subspace. We further show that the reduced trajectories are differentiable with respect to the subspace bases over the entire Grassmannian manifold, under proper choices of timestep sizes and numerical integrators. Finally, we propose a loss function measuring the trajectory-wise discrepancy between the original and reduced models. By tensor precomputation, we show that gradient information of such loss function can be evaluated efficiently over a long trajectory without time-integrating the high-dimensional dynamic system. Through evaluations on a row of simulation benchmarks, we show that our method reduces the discrepancy by $50\% - 90\%$ over conventional reduced models.

## 1 INTRODUCTION

High-dimensional Partial Differential Equations (PDE), especially fluid dynamic systems, find vast applications in the field of scientific computation Moin & Mahesh (1998); Alfonsi (2009), PDE-constrained optimization Biegler et al. (2003); Herzog & Kunisch (2010), design prototyping Baysal & Eleshaky (1992); Zang & Green (1999), fluidic devices design Du et al. (2020); Li et al. (2022), and digital entertainment Bridson & Batty (2010); Bridson (2015), to name a few. A fundamental task of all these applications lies in the efficient prediction of numerical solutions over a long horizon. In design prototyping, for example, a designer needs to quickly preview the fluid flow surrounding an aerial vehicle in order to refine its form factor. In a game engine, a fluid simulator needs to achieve real-time performance to provide interactive special effects for players. Although abundant numerical tools Petrila & Trif (2004); Demkowicz et al. (1989) have been developed over the past decades with improved efficacy, their algorithmic complexity is still challenging the limits of current computational resources. As a parallel effort, the idealized, incompressible, inviscid Eulerian fluid should be time reversible and energy preserving Duponcheel et al. (2008), and dedicated numerical schemes are proposed to faithfully preserve these properties in a discrete setting Rowley & Marsden (2002); Pavlov et al. (2011). This implies that the initial condition of a trajectory can be recovered from any state thereafter and the discrete total energy is a constant throughout the predicted trajectory. Although idealized fluid models are not pursued in applications, their accurate prediction is an important criterion of reliable numerical schemes.

Since their proposal Berkooz et al. (1993); Rowley (2005), model reduction has been quickly established as one of the most effective approaches that can significantly reduce the PDE prediction cost. By restricting the state variables to low-dimensional linear and nonlinear sub-manifolds, the dimension of associated dynamic system can be reduced by orders of magnitude. Over the years, several data-driven and data-free approaches have been proposed to identify sub-manifolds that can capture the complex dynamic behaviors of fluids. The earliest data-driven method of Proper Orthogonal

Decomposition (POD) Berkooz et al. (1993) finds the optimal linear subspace that best explains the variation of the state distribution. However, POD is flawed in that it ignores the temporal dependence of state variables. This problem is remedied by the Dynamic Model Decomposition (DMD) Schmid (2010) that finds the optimal linear subspace that best approximates the Koopman operator. However, these data-driven algorithms are irrespective of the nonlinearity in the underlying PDE. Comparatively, data-free methods, such as balanced POD Rowley (2005), $\mathcal{H}_2$-optimization Gugercin et al. (2006), and modal analysis Taira et al. (2017), identify bases corresponding to the intrinsic property of PDE by analyzing the system transfer matrices in the frequency domain, and are thus independent of data. Unfortunately, these techniques are largely limited to linear systems and their extensions to nonlinear fluid dynamics, such as Yang et al. (2019), are in their infancy.

More generally, the construction of reduced fluid models has been formulated as machine learning problems for system identification. The vast majority of prior works generalize the non-intrusive approach and identify the state transfer function via supervised learning in an existing sub-manifold, where the transfer functions are parameterized via radial basis functions Zhang et al. (2016), feed-forward networks Hsieh & Tang (1998), recurrent networks Pearlmutter (1989); Wang et al. (2018), etc. More recent approaches jointly learn the state transfer function and identify the sub-manifold via convolutional autoencoder Wu et al. (2021); Hasegawa et al. (2020). Unfortunately, all these non-intrusive learning techniques cannot preserve the time reversible property of idealized fluid, potentially leading to large prediction error or requiring a large dataset.

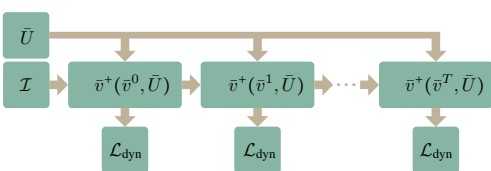

Figure 1: Given a distribution of initial conditions $\mathcal{I}$, we identify a reduced-order fluid model $\bar{v}^+(\bar{v}, \bar{U})$ by optimizing the bases $\bar{U}$ that minimize the expected trajectory-wise discrepancy loss $\mathcal{L}_{\mathrm{dyn}}$. Our output model $\bar{v}^+(\bar{v}, \bar{U})$ can perform efficient and as-accurate-as-possible fluid trajectory predictions.

We propose a machine learning approach to identify locally optimal, time reversible, reduced-order fluid dynamic models. We first interpret the linear subspace of fluid velocities as a point on the Grassmannian manifold and study the dependence of reduced trajectories on the choice of subspace. Thanks to the time reversibility, we show that the map from the subspace bases to reduced trajectories is globally differentiable, which allows us to optimize the reduced model via gradient-based Riemannian optimization. We further propose a trajectory-wise discrepancy loss that penalizes the difference between the full-order and the reduced trajectories. To make the optimization tractable, we propose a tensor precomputation scheme to accelerate the back-propagation of gradient information. Figure 1 illustrates the high-level pipeline of our method that fine-tunes the reduced fluid model to minimize the expected trajectory-wise discrepancy loss over the distribution of initial conditions. In essence, our method extends prior optimal reduced bases construction algorithm Berkooz et al. (1993); Schmid (2010) to the nonlinear, idealized fluid dynamic model. As an intrusive approach, our method preserves the desirable property of time reversibility. When compared with POD-type reduced model baseline on a row of idealized fluid simulation benchmarks, our method lowers the discrepancy by $50\% - 90\%$.

## 2 RELATED WORK

We review related works on machine learning for solving ODE and PDE, reduced physics models beyond fluid dynamics, and finally learning under hard constraints.

**Learning for Solving ODE and PDE:** To study the complex behavior of dynamic systems, various mathematical models have been proposed for idealized models of fluid, solid, elasto-magnetic fields, etc. However, there are oftentimes subtle discrepancies between these models and real-world observations that are hard to model, in which cases machine learning stands out as an effective approach for acquiring these behaviors from groundtruth data. Chen et al. (2018) propose to learn such dynamics as a general Ordinary Differential Equation (ODE) with the time derivative of state predicted via a neural network. Although this method is applicable to general dynamic systems, it does not reflect the spatial and temporal structures of certain systems, which limits its accuracy, data-efficacy, and scalability to high-dimensional systems such as fluids. Several follow-up works improve the network architecture to reflect additional structures. For example, the inter-dependency

among spatial variables is oftentimes local and sparse, which could be modeled via neighborhood message passing Battaglia et al. (2016); Li et al. (2019). Hamiltonian dynamics are time reversible and energy preserving, which is modeled by learning the Hamiltonian operator in "canonical" coordinates Greydanus et al. (2019), generalized coordinates Cranmer et al. (2020), or ambient space with additional constraints Finzi et al. (2020). **However, the above techniques are using Lagrangian coordinates, while fluid mechanics are oftentimes modeled via an Eulerian grid, see e.g. Takahashi et al. (2021), which is a major point of difference from our method. Parallel efforts have been made to learn Eulerian fluid mechanics Um et al. (2020); Takahashi et al. (2021); Holl et al. (2020); Prantl et al. (2019); Kim et al. (2019). Some of these works Um et al. (2020); Takahashi et al. (2021); Holl et al. (2020) learn to control fluids via differentiable simulators but the dynamic systems are not learned. Other works Prantl et al. (2019); Kim et al. (2019) learn to predict short future trajectories of free-surface flows. As the major difference from these techniques, our goal is to predict arbitrarily long trajectories by utilizing the time reversible structure of the dynamic system to guarantee stability. On the downside, however, our method cannot predict free-surface flows.**

**Learning Reduced Physical Models:** Model reduction is a special kind of dimension reduction technique dealing with time series datasets and we refer readers to Rowley & Dawson (2017) for a review of its application in fluid mechanics. Other than fluid, reduced models have been adopted in predicting the behaviors of solid Sampaio & Soize (2007), electromagnetic fields Ralph-Uwe et al. (2008), quantum and molecular mechanics Mohan & Fredrickson (2020), neuron propagations Amsallem & Nordstrom (2016), etc. A successful reduced model involves two steps: 1) embed the data into a proper subspace that well explains the data variations; 2) project the dynamic system into the subspace. Conventional techniques for model reduction are restricted to linear dynamic systems, for which optimal linear subspace can be identified via POD or DMD Berkooz et al. (1993); Rowley (2005) and the projected dynamic system can be precomputed via Galerkin projection. More general machine learning techniques have been proposed for an extension to nonlinear dynamics. For example, convolution autoencoder has been used to identify nonlinear subspaces Wu et al. (2021); Hasegawa et al. (2020). The ROM-net Daniel et al. (2020) learns to select a suitable subspace from a dictionary. Li et al. (2017) proposes to represent the linear subspace bases as the output of a universal neural network. In order to efficiently project the nonlinear dynamic system into the subspace, the Discrete Empirical Interpolation Method (DEIM) Chaturantabut & Sorensen (2010) has been proposed to select a sparse set of interpolation points. The interpolation points are then contracted with the subspace bases in an intrusive manner. Non-intrusive approaches use universal neural networks to learn the entire nonlinear transfer function Wu et al. (2021); Hasegawa et al. (2020); Lee et al. (2021) or part of the nonlinear terms Maulik et al. (2019). It has been noticed in Amsallem & Nordstrom (2016); Liu et al. (2015) that time reversibility and energy preservation features can be preserved by using an intrusive approach, which is a main reason behind our technical choice.

**Learning Under Hard Constraints:** Our work deals with idealized fluid satisfying two hard constraints: 1) incompressibility and 2) time reversibility. Since prominent training algorithms Duchi et al. (2011); Kingma & Ba (2014) and neural network architectures are designed for unconstrained optimization, dealing with hard constraints has been a long-standing problem Márquez-Neila et al. (2017). There are two general approaches to inform a learned model of hard constraints: softening and constraint layers. Softening transforms the hard constraint into soft losses and relies on unconstrained optimizations. Some hard constraints model invariant variables, in which case data augmentation could be used to enforce a neural network gives the same output over all invariant transforms of inputs. In the learning of physical models, softening has been adopted to enforce physical correctness Sirignano & Spiliopoulos (2018); Ober-Blöbaum & Offen (2022), fluid incompressible Ajuria Illarramendi et al. (2020), and collision-free constraints Tan et al. (2022), and data augmentation has been used to enforce invariance to rigid Morozov et al. (2021) and Galilean transformations Ling et al. (2016). A common problem with all these approaches lies in the unpredictable constraint violation in regions of insufficient data coverage. To exactly impose hard constraints, a series of works Amos & Kolter (2017); Agrawal et al. (2019) propose to formulate the constrained optimization as a differentiable layer in the neural network architecture. In particular, the entire fluid simulator has been formulated as a differentiable layer Schenck & Fox (2018); Takahashi et al. (2021) for model-based control and system identification. The incompressible constraint has also been formulated as an elliptic PDE solver layer in Mohan et al. (2020). Although these techniques can enforce hard constraints, the cost of forward- and back-propagations through these layers are

prohibitive. Even worse, these layers must be evaluated during both training and test time. Our method uses the constraint layer approach to enforce fluid incompressibility and time reversibility, by incorporating the reduced model Liu et al. (2015) as our differentiable layer. However, we encode the constraint property into the reduced bases, which is fixed during test time, leading to the low computational cost of trajectory prediction.

## 3 TIME REVERSIBLE REDUCED FLUID MODEL

We briefly review the underlying geometric structure and associated computational model of idealized, incompressible, inviscid fluid Pavlov et al. (2011). Given a simulation domain $\mathcal{M}$, the fluid configuration can be described as a vector field $v \in \mathcal{V}(\mathcal{M})$ where $v(x)$ for any $x \in \mathcal{M}$ represents the velocity of fluid at $x$. The governing equation for $v$ is:

$$\dot{v} + \nabla \times v \times v + \nabla \lambda = 0 \qquad \text{s.t. } \nabla \cdot v = 0, \tag{1}$$

where $\lambda$ is the pressure field, which is also the Lagrangian multiplier for the divergence-free constraint $\nabla \cdot v = 0$. The above system is closed with appropriate initial and boundary conditions. Pavlov et al. (2011) proposed time-reversible, energy preserving spatial and temporal discretization schemes for Equation 1. However, directly time integrating the discrete system requires solving large-scale nonlinear system equations. Reduced-order model Liu et al. (2015) scales down the cost by embedding $v$ into a $p$-dimensional subspace with divergence-free, orthogonal basis $U$, giving $v = Uz$ where $z$ is the coefficient vector. The reduced-order governing equation can be derived via Galerkin projection:

$$\dot{z} + \int_{\mathcal{M}} U^T \nabla \times (Uz) \times (Uz) = 0, \tag{2}$$

where the second term is the reduced-order advector, which could be succinctly written as a contraction with a third-order tensor $C_{kij}$:

$$\dot{z}_k + \sum_i \sum_j C_{kij} z_i z_j = 0 \qquad \text{s.t. } C_{kij} \triangleq \int_{\mathcal{M}} \langle U_k, \nabla \times U_i \times U_j \rangle, \tag{3}$$

where we use $z_k$ (resp. $U_k$) to denote the $k$th element (resp. column). For fast reduced trajectory prediction, the tensor $C_{kij}$ is precomputed, and a small $p$ is used. An essential feature of $C_{kij}$ is antisymmetry: $C_{kij} = -C_{jik}$, which implies that the continuous-time, reduced system is also energy-preserving as:

$$\frac{d}{dt}\|z\|^2 = 2\sum_{kij} C_{kij} z_k z_i z_j = \sum_{kij}(C_{kij} - C_{jik}) z_k z_i z_j = 0.$$

Using a variational integrator, e.g. the trapezoidal rule, the energy will also be conserved in a time-discrete computational model. We use a superscript $^+$ to denote variables at the next time instance, the superscript $^d$ denotes the variable at the $d$th timestep, and $\delta t$ denotes timestep size. The trapezoidal rule relates $z^+$ and $z$ by:

$$z^+(z): \frac{z^+ - z}{\delta t} + \mathbb{C}(z^+) = 0 \qquad \text{s.t. } \mathbb{C}(z^+) \triangleq \sum_{ij} C_{:ij} \frac{z_i^+ + z_i}{2} \frac{z_j^+ + z_j}{2}, \tag{4}$$

from which $z^+$ can be solved via the Newton-Raphson method to satisfy $\|z^+\|^2 = \|z\|^2$, i.e. energy conservation, as well as discrete-time reversibility. These remarked features, originally discovered in Pavlov et al. (2011); Liu et al. (2015), achieve an ideal balance between computational efficacy and numerical stability. **As pointed out by Pavlov et al. (2011), although real-world flows are not ideally energy-preserved, simulating ideal flows is a crucial benchmark for evaluating the stability and fidelity of a simulator. More general non-reversible flows can be modeled by adding additional constitutive terms. As an example, we could add a viscous term $\mu \nabla (\nabla \cdot v)$ to model energy dissipation and this term can be projected to the reduced space via Galerkin projection. In Figure 2, we plot the procedure of energy dissipation under different $\mu$ using both our learned reduced model and the groundtruth fullspace model.** We formalize and prove these properties in Appendix A and Appendix B. In particular, Equation 4 defines a unique $z^+$ given $z$ and a sufficiently small $\delta t$, so we define the function $z^+(z)$ by a slight abuse of notations. The accuracy of a reduced model relies on a proper choice of the basis vector $U$, which remains a difficult but underappreciated problem.

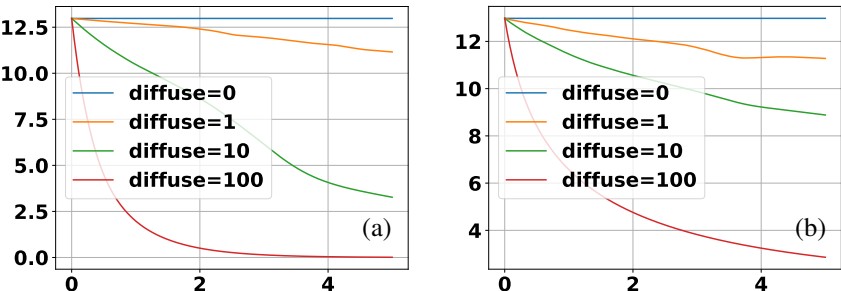

Figure 2: **We plot the energy dissipation cause by a viscous term under** $\mu = 0, 1, 10, 100$, **simulated using our learned reduced model (a) and the groundtruth fullspace model (b).**

## 4 REDUCED MODEL OPTIMIZATION

As illustrated in Figure 1, we propose to identify reduce-order fluid models via gradient-based optimization of $U$ to minimize the trajectory-wise discrepancy between a reduced-order model (Equation 3) and the full-order model (Equation 1). In this section, we first discretize the spatial computational domain (Section 4.1), we then propose our discrepancy loss function (Section 4.3), and finally discuss our optimization algorithm (Section 4.4).

### 4.1 SPATIAL DISCRETIZATION

We assume that $\mathcal{M}$ is discretized using a tetrahedron mesh or a rectangular grid via Discrete Exterior Calculus (DEC) as in Pavlov et al. (2011). As a result, each vector field has a finite dimension $n \gg p$. We use a bar to denote discrete variable so $\bar{v} \in \mathbb{R}^n$. $\bar{U}$ belongs to the intersection of Stiefel manifold $\text{St}(n, p)$ and the divergence-free basis subspace: $\mathcal{D}(n, p) = \{\bar{U} \in \mathbb{R}^{n \times p} | \bar{\nabla} \cdot \bar{U} = 0\}$, where $\bar{\nabla} \cdot \in \mathbb{R}^{(n-m) \times n}$ is the discrete divergence operator and $m \gg p$ is the dimension of divergence-free velocity subspace. The elements of $\bar{U}$ can also be identified with the elements of $\text{St}(m, p)$. Indeed, we can find a set of unit, orthogonal bases $\bar{D} \in \mathbb{R}^{n \times m}$ spanning the subspace of divergence-free velocity fields. For each $\bar{U}$, we can identify some $\bar{U}^m \in \text{St}(m, p)$ such that $\bar{U} = \bar{D}\bar{U}^m$. As illustrated in Figure 8, a point on $\text{St}(n, p)$ is the bases of a $p$-dimensional velocity field subspace, while a point on $\text{St}(m, p)$ is the bases of a $p$-dimensional *divergence-free* velocity field subspace. Since we merely use $\bar{U}$ to project the velocity field into a subspace, we are only interested in the lower-dimensional Grassmannian Manifold (the manifold of velocity subspace irrespective of the particular bases), but we use Stiefel representation for better memory and computational efficacy. In other words, we treat $\bar{U}$ as our decision variable. We further write the tensor coefficient $C_{kij}$ as a function $C(\bar{U}_k, \bar{U}_i, \bar{U}_j)$, which is derived by discretizing the continuous definition of $C_{kij}$ in Equation 3 using DEC.

### 4.2 LIFTING TRANSFER FUNCTION FROM REDUCED- TO FULL-SPACE

In order to optimize the accuracy of reduced dynamic system, we first need to compare simulated trajectories generated by different bases $\bar{U}$. However, the coordinate vector $z$ of different $\bar{U}$ is incomparable, as they reside in different linear subspaces. To resolve this problem, we propose to lift $z$ to $\bar{v} = \bar{U}z$ in the ambient space $\mathbb{R}^n$, so that two vectors can be compared by the induced metric in the Euclidean space. Further, we can smoothly extend the reduced-order simulator function to the ambient space using the projection operator $\bar{P} = \bar{U}\bar{U}^T$ and $\bar{P}_\perp = I - \bar{P}$:

$$\bar{v}^+(\bar{v}, \bar{U}) \triangleq \bar{U}z^+(\bar{U}^T\bar{v}) + \bar{P}_\perp\bar{v}. \tag{5}$$

In other words, the velocity component orthogonal to the subspace is stationary, and the tangential velocity is governed by the reduced dynamic system. As detailed in Appendix C, the above extension can be written as a function defined on the Grassmannian manifold: $\bar{v}^+(\bar{v}, \bar{P}^m) : \mathbb{R}^n \times \text{Gr}(m, p) \mapsto \mathbb{R}^n$, where we denote $\bar{P}^m = \bar{U}^m [\bar{U}^m]^T$. With the smooth extension, we can evaluate the derivatives of $\bar{v}^+$ with respect to $\bar{v}$ and the subspace. We can also compare two velocity fields generated by reduced-order simulators using different subspaces. Note the full-order dynamics (Equation 1) can be identified with $U^m = I^{m \times m}$. The above lifting is not unique, and a useful alternative is to discard the orthogonal component, i.e. setting $\bar{P}_\perp\bar{v}^+ = 0$, which is discussed in Appendix C.2. As our major

contribution, we show in Appendix C that the above function $\bar{v}^+$ is a well-defined smooth function on $\mathrm{Gr}(m, p)$. We further show that for any differentiable loss function $\mathcal{L}(\bar{v}^+)$, its derivatives with respect to the bases can be efficiently computed under a proper representation of $\bar{U}$ as a manifold point.

---

**Algorithm 1** **Forward-Backward**$(\bar{v}^0, \bar{U})$

---

1: Precompute tensor $C_{kij} = C(\bar{U}_k, \bar{U}_i, \bar{U}_j)$
2: Precompute tensor $C(\bar{D}\bar{D}^T, \bar{U}_i, \bar{U}_j)$
3: **for** $d = 0, \cdots, T - 1$ **do**            ▷ forward propagation
4:     $\bar{v}^{d+1} \leftarrow \bar{v}^+(\bar{v}^d, \bar{U})$
5: $G \leftarrow 0$                                ▷ backward propagation
6: Evaluate $\nabla \mathcal{L} \leftarrow \frac{\partial \gamma^T \mathcal{L}_{\mathrm{dyn}}(\bar{v}^T)}{\partial \bar{v}^T}$
7: **for** $d = T - 1, \cdots, 1$ **do**
8:     $G \leftarrow G + $ Equation 12
9:     $\nabla \mathcal{L} \leftarrow {\frac{\partial \bar{v}^{d+1}}{\partial \bar{v}^d}}^T \nabla \mathcal{L} + \frac{\partial \gamma^d \mathcal{L}_{\mathrm{dyn}}(\bar{v}^d)}{\partial \bar{v}^d}$
10: Compute $\nabla_{\bar{U}} \mathcal{L}$ via Equation 11          ▷ divergence-free projection
11: Return $\nabla_{\bar{U}} \mathcal{L}$

---

**Algorithm 2** **RAMSGRAD**$(\mathcal{I}, \bar{U})$

---

**Input:** $\beta_1, \beta_2, \alpha, \delta t$
1: $m \leftarrow 0, \tau \leftarrow 0, \nu \leftarrow 0, \hat{\nu} \leftarrow 0$
2: **while** Not converge **do**
3:     Sample $v^0 \sim \mathcal{I}$               ▷ we always use batch size equals to 1
4:     $g \leftarrow$ Forward-Backward$(v^0, \bar{U})$
5:     $m \leftarrow \beta_1 \tau + (1 - \beta_1) g$
6:     $\nu \leftarrow \beta_2 \nu + (1 - \beta_2) \|g\|^2$
7:     $\hat{\nu} = \max(\hat{\nu}, \nu)$
8:     $\bar{U} \leftarrow$ Retract$(\bar{U}, -\alpha m / \sqrt{\hat{\nu}})$         ▷ by QR factorization
9:     $\tau \leftarrow \bar{P}_\perp m$                 ▷ approximate parallel transport
10: Return $\bar{U}$

---

### 4.3 REDUCED DISCREPANCY LOSS

The differentiable structure of reduced fluid allows us to minimize the discrepancy between reduced- and full-order model in an efficient model-based manner. Given two velocity fields $\bar{v}$ and $\bar{v}^+$, a full-order model should satisfy the governing equation of motion, which inspires the following discrepancy measure:

$$\mathcal{L}_{\mathrm{dyn}}(\bar{v}^+, \bar{v}) \triangleq \left\| \bar{D}\bar{D}^T \frac{\bar{v}^+ - \bar{v}}{\delta t} + C(\bar{D}\bar{D}^T, \frac{\bar{v}^+ + \bar{v}}{2}, \frac{\bar{v}^+ + \bar{v}}{2}) \right\|^2. \tag{6}$$

This is similar to the physics correctness loss used in Sirignano & Spiliopoulos (2018); Ober-Blöbaum & Offen (2022) and we absorb the linear divergence-free constraint by using the projection operator $\bar{D}\bar{D}^T$. Again, evaluating $\mathcal{L}$ involves a sparse linear solve for each of the $T$ timesteps. But we can accelerate this computation thanks to the low-rank property of the velocity fields. Since, $\bar{v}$ and $\bar{v}^+$ both reside in low-rank spaces, we can write:

$$C(\bar{D}\bar{D}^T, \frac{\bar{v}^+ + \bar{v}}{2}, \frac{\bar{v}^+ + \bar{v}}{2}) = \sum_{ij} C(\bar{D}\bar{D}^T, \bar{U}_i, \bar{U}_j) \frac{\bar{z}_i^+ + \bar{z}_i}{2} \frac{\bar{v}_j^+ + \bar{v}_j}{2},$$

and precompute the tensor $C(\bar{D}\bar{D}^T, \bar{U}_i, \bar{U}_j)$ via $p^2$ sparse linear solves at the cost of $\mathcal{O}(n^\omega p^2)$. For a trajectory with $T \gg p^2$ timesteps, this operator reduces the cost of evaluating $\mathcal{L}_{\mathrm{dyn}}$ from $\mathcal{O}(n^\omega T)$ to $\mathcal{O}(n^\omega p^2 + Tn p^2)$.

### 4.4 STOCHASTIC RIEMANN OPTIMIZATION

Using a low-dimensional subspace, it is impossible to approximate all fluid simulation trajectories with sufficient accuracy. Instead, reduced models are designed to optimize a subset of trajectories with a given distribution $\mathcal{I}$ of initial conditions, i.e. $\bar{v}^0 \sim \mathcal{I}$ and our goal is to solve the following problem via stochastic Riemann optimization:

$$\underset{\bar{U} \in \mathcal{D}(n,p) \cap \mathrm{St}(n,p)}{\operatorname{argmin}} \quad \mathbb{E}_{\bar{v}^0 \sim \mathcal{I}}\left[\sum_{d=1}^{T} \gamma^d \mathcal{L}_{\mathrm{dyn}}(\bar{v}^d, \bar{v}^{d-1})\right], \tag{7}$$

where $T$ is the horizon of trajectory and $\gamma \in (0,1]$ is a constant discount factor. Riemann optimization is a well-studied problem in both deterministic and stochastic settings and we use the RAMSGRAD algorithm proposed in Becigneul & Ganea (2019). This algorithm requires both the retraction and parallel transport operators on $\mathrm{St}(n,p)$. We use QR-factorization for the retraction operator Bendokat et al. (2020). Unfortunately, there is no efficient way to compute the parallel transport operator Edelman et al. (1998), so we approximate the transport operator by projecting out the non-tangential component. This corresponds to using a single step of forward Euler integrator to solve the associated ODE of the transport operator. Again due to time reversibility, the objective function is globally differentiable with respect to $\bar{U}$ under compact $\mathcal{I}$ and sufficiently small $\delta t$. We outline our forward-backward gradient propagation in Algorithm 1 and adapted RAMSGRAD in Algorithm 2. These algorithms are well-defined due to the following lemma:

**Lemma 4.1.** *For any compact initial distribution $\mathcal{I}$, there exists a sufficiently small $\delta t$, such that the objective function $\sum_{d=0}^{T} \gamma^d \mathcal{L}_{dyn}(\bar{v}^d)$ is globally differentiable, i.e. for any $z^0 \in \mathcal{I}$ and $\bar{U} \in \mathcal{D}(n,p) \cap St(n,p)$.*

*Proof.* Since $\mathcal{I}$ is compact, $\bar{v}^0$ is uniformly upper bounded by some $r$ and $\|z^0\| = \|\bar{U}^T \bar{v}^0\| \leq r$. By Corollary A.5, there exists a sufficiently small $\delta t$ making any $z^d$ a differentiable, reversible function of $z^0$. This also implies $\bar{v}^d$ is a differentiable, reversible function of $\bar{v}^0$ under the definition of Equation 5, and our result follows. $\square$

## 5 EVALUATION

We implement our method using Pytorch with a fluid simulator implemented via native C++ with CPU parallelization, and perform all the computations on an AMD Threadripper 3970X CPU having 32 cores. We initialize our method using a conventional POD-type algorithm. Given $\mathcal{I}$, we first sample a set of $N$ trajectories using the full-order dynamics (Equation 1) and then perform a POD-type basis extraction. The number of extracted bases is determined by truncating the eigenvalues below $\epsilon$ of the largest eigenvalue. We always use a batch size of 1. The performance of our method is summarized in Table 1. We consider two variants of our method:

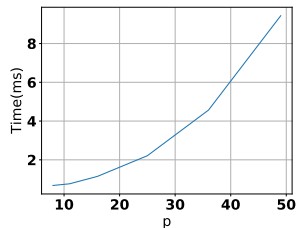

Figure 3: The cost of evaluating $z^+(z)$ plotted against $p$.

coupled case, where $C_{kij}$ is treated as a function $C(U_k, U_i, U_j)$ as discussed in Section 4, and decoupled case, where $C_{kij}$ is treated as an antisymmetric independent decision variable. Our main experiments are performed with the coupled case. Experiments with the decoupled case and a summary of decision variables are included in Appendix D. The efficacy of trajectory prediction using a reduced-order model depends on $p$ as illustrated in Figure 3, so the runtime performance of both the POD baseline and our method are the same, while the cost of evaluating the full-order model is 252ms (26× slower than the reduced-order model with $p = 49$).

| Benchmark | $\epsilon = 0.05$ | | | $\epsilon = 0.01$ | | | $\epsilon = 0.001$ | | | $\epsilon = 0.0001$ | | |
|---|---|---|---|---|---|---|---|---|---|---|---|---|
| | $p$ | Loss-POD | **Loss-Ours** | $p$ | Loss-POD | **Loss-Ours** | $p$ | Loss-POD | **Loss-Ours** | $p$ | Loss-POD | **Loss-Ours** |
| Taylor Vortices | 8 | 4.84 | 0.58 | 11 | 3.93 | 0.39 | 16 | 1.99 | 0.17 | 25 | 0.70 | 0.07 |
| Plume Rise | 6 | 57.04 | 6.37 | 9 | 28.23 | 5.38 | 15 | 18.53 | 2.30 | 26 | 5.96 | 1.44 |
| Plume Rise+Obstacle | 5 | 133.16 | 10.48 | 8 | 46.47 | 8.60 | 16 | 17.57 | 2.57 | 30 | 5.71 | 1.05 |
| Spherical Plume | - | - | - | 36 | 120.89 | 44.89 | - | - | - | - | - | - |
| Two Plume | - | - | - | 59 | 103.23 | 49.22 | - | - | - | - | - | - |

Table 1: Summary of benchmarks for comparing POD and our method under different $\epsilon$ and $p$.

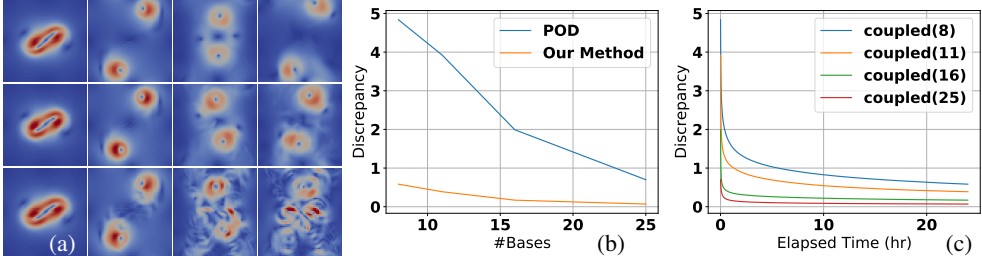

Figure 4: (a) Velocity magnitude field snapshots of the Taylor vortices benchmark, generated by full-order model (top row), our method with $\epsilon = 0.0001$ and $p = 25$ (middle row), and POD with $\epsilon = 0.0001$ and $p = 25$ (bottom row). (b) Trajectory-wise discrepancy loss of POD and our method, under different $p$. (c) The convergence history of our method over 24 hours.

Our first benchmark is Taylor vortices Pavlov et al. (2011), where two vortices are separated by a distance slightly larger than the critical threshold. We use a velocity field discretized on a $64 \times 64$ rectangular grid with the periodic boundary condition, leading to $n = 8192$. This is a single trajectory ($\mathcal{I}$ is deterministic) and we set $T = 500$, $\delta t = 0.01$. We experiment with four parameters $\epsilon = 0.05$, $0.01$, $0.001$, and $0.0001$ and the number of bases is $p = 8$, $11$, $16$, and $25$, correspondingly. With each $\bar{U}$ as the initial guess, we run our optimizer for 24 hours. In Figure 4bc, we plot the trajectory-wise discrepancy loss against the number of bases $p$ and the convergence history of our method. Compared with POD bases, our method reduces the discrepancy loss by $87.93\%$, $90.12\%$, $91.47\%$, and $90.16\%$, respectively. Snapshots are shown in Figure 4a, where our method predicts a velocity field closer to the full-order groundtruth.

Our second benchmark involves having a smoke plume rise at a constant speed. We use a rectangular domain of $[0, 1]^2$ with all Dirichlet boundary conditions. The region of $[0.25, 0.75] \times [0.125, 0.375]$ is occupied by the smoke with a constant speed $(0, 1)$, the remaining regions have zero velocity, and we use $T = 1000$. All other settings are the same as our first benchmark. The discrepancy loss and convergence history are plotted in Figure 13bc. We experiment with four parameters $\epsilon = 0.05$, $0.01$, $0.001$, and $0.0001$, the corresponding numbers of bases $p$ are $6$, $9$, $15$, and $26$, respectively. Our method reduces the discrepancy loss by $88.82\%$, $80.94\%$, $87.60\%$, and $75.79\%$, respectively. We have also tested a variant of our method with an obstacle in the simulation domain, where our method reduces the discrepancy loss by $92.13\%$, $81.49\%$, $85.38\%$, and $81.70\%$, respectively. Snapshots of our second benchmark are shown in Figure 13a and Figure 14 of Appendix H.

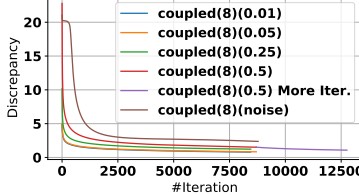

| $\tilde{\epsilon}$ | Loss-Init. | Loss-8k-Iter. | Loss-12k-Iter. |
|---|---|---|---|
| 0.01 | 4.84 | 0.79 | N/A |
| 0.05 | 5.04 | 0.83 | N/A |
| 0.25 | 10.16 | 1.23 | N/A |
| 0.5 | 22.77 | 1.51 | 1.08 |
| Noise | 21.06 | 2.37 | N/A |

Figure 5: The convergence history of four instances of learning reduced Taylor vortices with $\epsilon = 0.05$, $p = 8$, and difference noise levels $\tilde{\epsilon} = 0.01, 0.05, 0.25, 0.5$. We first run the four training instances for 8000 iterations, which already brings the ultimate discrepancy loss down to similarly low levels. We then give the $\tilde{\epsilon} = 0.5$ instance another 4500 iterations (purple after red curve) and it could outperform the $\tilde{\epsilon} = 0.25$ instance. Finally, we tried using a fully noisy initialization of $\bar{U}$ and the result is much worse than other instances.

In our first benchmark, Taylor vortices Pavlov et al. (2011), we further analyze the sensitivity of our method with respect to the initial guess $\bar{U}$. To this end, we first compute $\bar{U}$ via POD and then corrupt $\bar{U}$ using a random noise bases $\tilde{U}$ with each element sampled according to the truncated normal distribution with $\mu = 0$, $\sigma = 1$ and truncated to range $[-1, 1]$. We then use the following initial guess: $\text{Retract}(\tilde{U}, \bar{D}\bar{D}^T\tilde{U}\Sigma)$, where $\Sigma$ is a scaling diagonal matrix such that each column of $\tilde{U}\Sigma$ has $l_2$-norm equals to some $\tilde{\epsilon}$ and $\tilde{\epsilon}$ controls the magnitude of random noise. Here multiplying by $\bar{D}\bar{D}^T$ ensures that our noise is divergence-free. In Figure 5, we profile the convergence history with $\tilde{\epsilon} = 0.01, 0.05, 0.25, 0.5$. Although the noise can drastically change the initial discrepancy loss, all four instances can reduce the loss to similar levels after sufficiently many iterations. Our analysis also implies that the POD baseline provides a good initial guess of $\tilde{U}$, because a fully noisy initialization of $\bar{U}$ can lead to a worse result.

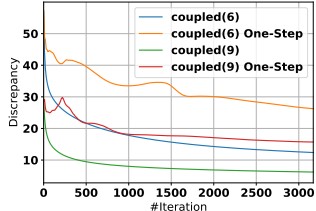

| Mode | $\epsilon/p$ | Loss-POD | Loss-3k-Iter. |
|------|------|----------|----------------|
| One-Step | 0.05/6 | 57.39 | 24.12 |
| Full-Unrolling | 0.05/6 | 57.39 | 6.37 |
| One-Step | 0.01/9 | 28.23 | 15.71 |
| Full-Unrolling | 0.01/9 | 28.23 | 5.38 |

Figure 6: The convergence history over 3000 iterations of four instances of learning reduced smoke plume rising trajectory. We use two sets of instances: $\epsilon = 0.05$, $p = 6$ and $\epsilon = 0.01$, $p = 9$. For each set, we compare one-step and full-unrolling mode of training.

In the recent work Brandstetter et al. (2022), authors proposed two training modes for learning neural PDE solver, one-step training and full-unrolling. One-step training cuts off the gradient after a single timestep, while the full-unrolling mode considers the full gradient of Equation 7 over the entire trajectory. We compare the two modes in Figure 6 in terms of trajectory-wise discrepancy loss, using our second benchmark scenario, rising smoke plume. Both modes can reduce the loss after 3000 iterations, although there is some initial fluctuation in one-step training, while full-unrolling leads to significantly faster convergence. We use the full-unrolling mode for all other examples.

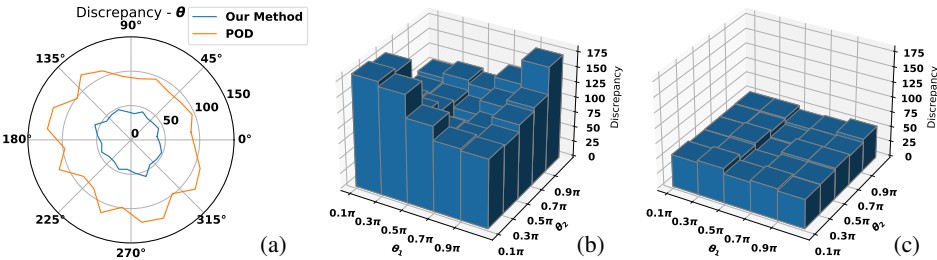

Figure 7: (a): The trajectory-wise discrepancy with respect to $\theta$ for our third benchmark. (bc): The initial (b) and final (c) trajectory-wise discrepancy with respect to $\theta_1, \theta_2$ for our forth benchmark.

Our third benchmark involves a spherical smoke plume, with initial diameter $1/3$ and speed $1.0$ located in the center of a $[0,1]^2$ domain, moving in varying directions. We assume the direction of motion is parameterized by the angle $\theta \in [0, 2\pi]$ sampled from the initial distribution $\mathcal{I} = \mathcal{U}([0, 2\pi])$. We use a velocity field discretized on a $64 \times 64$ rectangular grid with Dirichlet boundary condition ($n = 8320$). **Our training dataset for the POD baseline contains $N = 8$ trajectories with evenly sampled $\theta = 0°, 45°, 90°, \cdots$. With $T = 500, \delta t = 0.01, \epsilon = 0.01, p = 36$, we run our method for $12200$ iterations, taking $72$ hours to converge. We then test our method on another $24$ evenly sampled $\theta = 7.5°, 22.5°, 30°, \cdots$, which are not covered by the training dataset (some snapshots can be found in Figure 15 of Appendix H).** As plotted in Figure 7a, our method reduces the discrepancy by $54.65\%$ on average.

Our fourth benchmark extends the third one by involving two smoke plumes, located at $(0.5, 0.25)$ and $(0.5, 0.75)$. The directions of motion $\theta_1, \theta_2 \in [0, \pi]$ are sampled from the initial distribution $\mathcal{I} = \mathcal{U}([0, \pi]^2)$ and we set $\epsilon = 0.01, p = 59$. **Our training dataset for the POD baseline contains $N = 25$ trajectories with 5 evenly sampled $\theta_{1,2} = 0°, 72°, 144°, 216°, 288°$. Other parameters are the same as those of our third benchmark. We run our method for $18000$ iterations, taking $72$ hours to converge (some snapshots can be found in Figure 16 of Appendix H). Afterwards, we test our method on another $25$ evenly sampled $\theta_{1,2} = 36°, 108°, 180°, 252°, 324°$ that are not covered by the training dataset.** As plotted in Figure 7bc, our method reduces the discrepancy by $59.28\%$ on average.

## 6  CONCLUSION

We propose a model-based approach to fine-tune reduced fluid dynamic systems. Our main idea is to rely on the differentiable structure between the state transfer function and the linear subspace bases to minimize the expected trajectory-wise discrepancy loss, over a distribution of initial conditions. By evaluating several simulation benchmarks, we show that our method outperforms the POD baseline. As our major limitation, our trajectory prediction has sequential dependence and cannot exploit GPU parallelization. Even with our tensor precomputation technique, the training

still takes hours on a desktop machine, which is orders of magnitude slower than the simple POD or DMD method. In addition, our method uses a linear subspace with limited expressivity as compared with universal neural networks Wu et al. (2021); Hasegawa et al. (2020); Lee et al. (2021) used by non-intrusive model reduction techniques. We speculate that using neural networks to represent the reduced bases $\bar{U}$ is possible as done in Li et al. (2017), although the orthogonal and divergence-free constraints will be more difficult to enforce. Enforcing these constraints exactly as in Mohan et al. (2020) would compromise the efficacy of reduced time integration.

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

# A    DISCRETE ENERGY PRESERVATION

We prove that energy preservation and time reversibility hold in a time-discrete setting.

**Lemma A.1.** *The tensor $C_{kij}$ is antisymmetric.*

*Proof.* This follows from the definition of $C_{kij}$:

$$C_{kij} = \int_{\mathcal{M}} \langle U_k, \nabla \times U_i \times U_j \rangle = \int_{\mathcal{M}} U_k^T (\nabla U_i - \nabla U_i^T) U_j$$
$$= - \int_{\mathcal{M}} U_j^T (\nabla U_i - \nabla U_i^T) U_k = - \int_{\mathcal{M}} \langle U_j, \nabla \times U_i \times U_k \rangle = -C_{jik},$$

where we used elementary vector identity that $(\nabla \times A) \times B = B \cdot (\nabla A - \nabla A^T)$. $\square$

Using the antisymmetry of $C_{kij}$, we can show that trapezoidal rule is indeed energy preserving.

**Lemma A.2.** *For any $z^+$ satisfying the trapezoidal rule, $\|z^+\| = \|z\|$.*

*Proof.* Multiplying the lefthand side of Equation 4 by $z_k^+ + z_k$ and summing over $k$, we have:

$$\frac{\|z^+\|^2 - \|z\|^2}{\delta t} + 2 \sum_{kij} \left[ C_{kij} \frac{z_k^+ + z_k}{2} \frac{z_i^+ + z_i}{2} \frac{z_j^+ + z_j}{2} \right]$$
$$= \frac{\|z^+\|^2 - \|z\|^2}{\delta t} + \sum_{kij} \left[ (C_{kij} + C_{jik}) \frac{z_k^+ + z_k}{2} \frac{z_i^+ + z_i}{2} \frac{z_j^+ + z_j}{2} \right] = \frac{\|z^+\|^2 - \|z\|^2}{\delta t} = 0,$$

from which our result follows. $\square$

Next, we show that the trapezoidal integrator (Equation 4) must have a solution by a proper choice of sufficiently small $\delta t$.

**Lemma A.3.** *There exists a sufficiently small $\delta t$ such that Equation 4 can be solved for $z^+$ via the following negative gradient flow:*

$$f(z^+) \triangleq z^+ - z + \delta t \mathbb{C}(z^+) \quad \dot{z}^+ \triangleq -\nabla f(z^+)^T f(z^+)/2,$$

*with initial guess $z^+ = z$.*

*Proof.* We consider the Lyapunov candidate $V(z^+) \triangleq \|f(z^+)\|^2$ on the ball $\mathcal{B}_r(z) = \{z^+ | \|z^+ - z\| \le r\}$. The negative gradient flow satisfies:

$$\dot{V}(z^+) = -\|\nabla f(z^+)^T f(z^+)\|^2 = -\|(I + \delta t \nabla \mathbb{C}(z^+)^T) f(z^+)\|^2$$
$$= -V(z^+) - 2\delta t f(z^+)^T \nabla \mathbb{C}(z^+)^T f(z^+) - \delta t^2 \|\nabla \mathbb{C}(z^+)^T f(z^+)\|^2$$
$$\le -(1 - \delta t) V(z^+) + (\delta t - \delta t^2) \|\nabla \mathbb{C}(z^+)^T f(z^+)\|^2.$$

Now since the eigenvalue of a Hermitian matrix is a Lipschitz function of matrix entries Golub & Van Loan (2013), we must have:

$$\underline{\rho}(\|z\|, r) \le \rho(\nabla \mathbb{C}(z^+) \nabla \mathbb{C}(z^+)^T) \le \bar{\rho}(\|z\|, r),$$

for some $\underline{\rho}, \bar{\rho}$ and any $z^+ \in \mathcal{B}_r(z)$. Combining the above estimation, we have:

$$\dot{V}(z^+) \le -(1 - \delta t) V(z^+) + (\delta t - \delta t^2) \bar{\rho}(\|z\|, r) V(z^+).$$

Obviously, with sufficiently small $\delta t$, we have $\dot{V}(z^+) \le -\epsilon V(z^+)$ for some $\epsilon \in (0, 1)$ and $z^+ \in \mathcal{B}_r(z)$. Next, consider the boundary case $z^+ \in \partial \mathcal{B}_r(z)$, where we have:

$$V(z^+) - V(z) = r^2 + \delta t \mathbb{C}(z^+)^T (z^+ - z) + 2\delta t^2 \left[ \|\mathbb{C}(z^+)\|^2 - \|\mathbb{C}(z)\|^2 \right]$$
$$\ge (1 - \delta t) r^2 + (\delta t^2 - \delta t) \|\mathbb{C}(z^+)\|^2 - \delta t^2 \|\mathbb{C}(z)\|^2,$$

and we can choose sufficiently small $\delta t$ such that $V(z^+) > V(z)$ for all $z^+ \in \partial \mathcal{B}_r(z)$. Our result follows from the exponential stability condition Murray et al. (2017). $\square$

In practice, however, continuous gradient flow cannot be realized, but a similar argument as Lemma A.3 can be used to show that the Newton–Raphson method is guaranteed to converge when minimizing $V(z^+)$ under sufficiently small $\delta t$:

**Lemma A.4.** *There exists a sufficiently small $\delta t$ such that Equation 4 can be solved for $z^+$ via the Newton-Raphson method:*

$$z^{(d)} = z^{(d-1)} - \nabla f(z^{(d-1)})^{-1} f(z^{(d-1)}),$$

*with initial guess $z^{(0)} = z$. Here we use superscript with bracket to denote iteration index.*

*Proof.* Consider the reduction of Lyapunov candidate $V(z)$ after one iteration, we have:

$$V(z^{(d)}) = \|f(z^{(d-1)} - \nabla f(z^{(d-1)})^{-1} f(z^{(d-1)}))\|^2 = \sum_k \|\frac{\delta t}{2} f(z^{(d-1)})^T H_k(z^{(d-1)}) f(z^{(d-1)})\|^2$$

$$H_k(z^{(d-1)}) \triangleq \nabla f(z^{(d-1)})^{-T} \frac{C_{k::} + C_{k::}^T}{2} \nabla f(z^{(d-1)})^{-1}.$$

By a similar argument as in Lemma A.3, we can choose sufficiently small $\delta t$ such that:

$$\rho(H_k(z^{(d-1)})) \leq \bar{\rho}(\|z\|, r) \quad V(z^{(d)}) \leq \frac{p \delta t^2 \bar{\rho}(\|z\|, r)^2}{4} \|f(z^{(d-1)})\|^4,$$

as long as $z^{(d-1)} \in B_r(z)$. We can also choose sufficiently small $\delta t$ such that:

$$V(z^{(d)}) \leq \epsilon V(z^{(d-1)}) \quad \forall z^{(d-1)} \in B_r(z) \wedge \|f(z^{(d-1)})\| \leq 1, \tag{8}$$

for some $\epsilon \in (0, 1)$. Next, we consider the Hessian of $V(z^+)$:

$$\nabla^2 V(z^+) = [I - \delta t \nabla \mathbb{C}(z^+)^T][I - \delta t \nabla \mathbb{C}(z^+)] + \delta t \nabla^2 \mathbb{C}(z^+) f(z^+) \triangleq I + \mathcal{O}(\delta t) R(z^+),$$

where $R(z^+)$ is a smooth, symmetric matrix function. We can further choose sufficiently small $\delta t$ such that $V(z^+)$ is 1/2-strongly convex and for any $z^+ \in B_{3r}(z)/B_r(z)$:

$$V(z^+) - V(z) \geq r^2/2 + \nabla V(z)^T(z^+ - z) = r^2/2 + \delta t^2 \mathbb{C}(z)^T \nabla \mathbb{C}(z)(z^+ - z).$$

By the smallness of $\delta t$, we have:

$$\begin{cases} V(z^+) > V(z) & \forall z^+ \in B_{3r}(z)/B_r(z) \\ \|\nabla f(z^{(d-1)})^{-1} f(z^{(d-1)})\| \leq 2r & \forall z^{(d-1)} \in B_r(z) \wedge V(z^{(d-1)}) \leq \min(1, r^2/2) \end{cases} . \tag{9}$$

Combining Equation 8 and Equation 9, we have for small enough $\delta t$:

$$\begin{cases} z^{(d)} \in B_r(z) \\ V(z^{(d)}) \leq \epsilon V(z^{(d-1)}) \end{cases} \quad \forall z^{(d-1)} \in B_r(z) \wedge V(z^{(d-1)}) \leq \min(1, r^2/2).$$

Our result follows by choosing sufficiently small $\delta t$ such that $V(z^{(0)}) \leq \min(1, r^2/2)$ and invoke the discrete exponential stability condition Aitken & Schwartz (1994). $\square$

Note the choice of $\delta t$ is only dependent on $\|z\|$ and $r$, which can be used to show that the timestep size can be fixed throughout the trajectory for time reversible fluid systems:

**Corollary A.5.** *Given an initial condition $z^0$, an energy preserving discrete trajectory can be computed by repeatedly solving Equation 4 for $z^k$ using a fixed timestep size $\delta t$ via the Newton-Raphson method.*

*Proof.* This result can be derived by induction on two facts: 1) $\|z^k\| = \|z^{k-1}\|$ by Lemma A.2; 2) To solve for $z^k$, $\delta t$ can be determined as a function $\delta t(\|z^{k-1}\|, r)$ by Lemma A.4. $\square$

## B    DISCRETE TIME REVERSIBILITY

The above result guarantees energy preservation throughout the trajectory. We now move on to show time reversibility in the discrete setting:

**Lemma B.1.** *There exists a sufficiently small $\delta t$, such that for any $z \in B_r(0)$, the negative gradient flow Equation 4 defines a invertible map from $z$ to $z^+$.*

*Proof.* Following the same argument as in Lemma A.4, we can choose sufficiently small $\delta t$ such that $V(z^+)$ is strongly convex when restricted to $B_{2r}(0)$ and the map $z^+(z) = \underset{z^+}{\mathrm{argmin}} V(z^+)$ is well-defined and differentiable Still (2018). The derivative of function $z^+(z)$ can then be derived via the implicit function theorem as:

$$\nabla z^+(z) = -\left[I + \delta t \nabla \mathbb{C}(z^+)\right]^{-1}\left[I - \delta t \nabla \mathbb{C}(z^+)\right].$$

By Lemma A.2, we know that $z^+ \in B_r(0)$ as well. Again by the lipschitz continuity of singular values, we can choose sufficiently small $\delta t$ such that $\det(\nabla z^+(z)) \neq 0$ throughout $B_r(0)$ and our result follows by the inverse function theorem. $\qquad\square$

Lemma B.1 can also be extended to the entire trajectory via induction:

**Corollary B.2.** *Given an initial condition $z^0 \in B_r(0)$ for some $r$, an energy preserving discrete trajectory can be computed by repeated solving Equation 4 for $z^k$ using a fixed timestep size $\delta t$, such that the resulting map $z^k(z^0)$ is invertible.*

*Proof.* By induction on Lemma A.2 and Lemma B.1, we know that $z^k(z^{k-1})$ is invertible for any $k > 0$ and our result follows by composition of invertible functions. $\qquad\square$

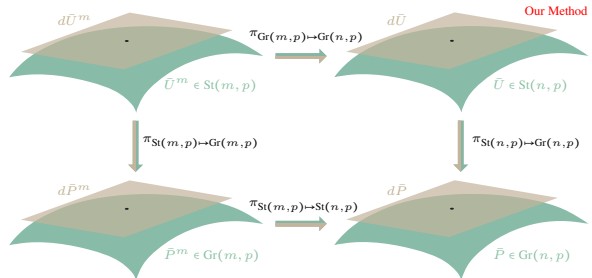

Figure 8: We illustrate the four manifolds: $\mathrm{St}(n, p)$ for the velocity bases; $\mathrm{St}(m, p)$ for the divergence-free velocity bases; $\mathrm{Gr}(n, p)$ for the velocity subspace; $\mathrm{Gr}(m, p)$ for the divergence-free velocity subspace. Our method maintains $\bar{U} \in \mathrm{St}(n, p)$ and represents the gradient as some $\nabla_{\bar{U}} \mathcal{L} \in \mathcal{T}_{\bar{U}} \mathrm{St}(n, p)$, which is both memory efficient and computationally tractable.

## C    DERIVATIVE FORMULATION

In this section, we analyze the differentiability of our lifted transfer function Equation 5. **To compute derivatives of the forward dynamic function with respect to the bases $\bar{U}$, we need to utilize the implicit function theorem and special representation of the bases as a manifold point, which cannot be exploited by automatic differentiation.** First, we show that the function is well-defined on the manifold $\mathrm{Gr}(m, p)$ via the following lemma:

**Lemma C.1.** *The lifted transfer function Equation 5 can be written as a function $\bar{v}^+(\bar{v}, \bar{P}^m)$.*

*Proof.* By the incompressibility of bases $\bar{U} = \bar{D}\bar{U}^m$, we have: $\bar{P} = \bar{D}\bar{P}^m\bar{D}^T$. Plugging this into Equation 5 and we have the follow rewrite:

$$\bar{v}^+(\bar{v}, \bar{P}^m) : \begin{cases} \bar{P}_\perp \bar{v}^+ = \bar{P}_\perp \bar{v} \\ \bar{P}\frac{\bar{v}^+ - \bar{v}}{\delta t} + C(\bar{P}, \bar{P}\frac{\bar{v}^+ + \bar{v}}{2}, \bar{P}\frac{\bar{v}^+ + \bar{v}}{2}) = 0 \end{cases},$$

from which our result follows. We can derive the original definition (Equation 5) by multiplying the second equation by $\bar{U}^T$ from the left. $\qquad\square$

Although the function is well-defined, the complexity of its derivative computation relies on an efficient representation of bases. A straightforward representation is to use matrix $\bar{P}^m$ and consider the function $\bar{v}^+(\bar{v}, \bar{P}^m)$. However, this representation requires storing the large matrix $\bar{P}^m$ which is computationally impractical. In this section, we exploit equivalent manifold representations to derive the computationally tractable formulas for the derivatives of arbitrary loss functions $\mathcal{L} \circ \bar{v}^+$. The relevant manifolds are illustrated in Figure 8. We first derive the partial derivative $\partial \bar{v}^+ / \partial \bar{v}$ via the implicit function theorem:

$$\frac{\partial \bar{v}^+}{\partial \bar{v}} = \left[ \bar{U} \left[ I + \delta t \nabla \mathbb{C}(z^+) \right]^{-1} \left[ I - \delta t \nabla \mathbb{C}(z^+) \right] \bar{U}^T + \bar{P}_\perp \right]. \tag{10}$$

The inverse of the system matrix above is well-defined when the timestep size $\delta t$ is sufficiently small according to Appendix A. It can be verified that the above derivative is invariant to the orthogonal basis transform. Next, we derive the partial derivative with respect to $\bar{P}^m \in \mathrm{Gr}(m, p)$. We denote $\bar{U}_\perp^m$ as the complement of $\bar{U}^m$ and $Q^m = \left( \bar{U}^m, \bar{U}_\perp^m \right) \in \mathrm{O}(m)$. An element of $\mathcal{T}_{\bar{P}^m} \mathrm{Gr}(m, p)$ can be identified with a matrix $dB \in \mathbb{R}^{(m-p) \times p}$ via:

$$d\bar{P}^m = Q^m \begin{pmatrix} & dB^T \\ dB & \end{pmatrix} [Q^m]^T .$$

We can lift $\mathrm{Gr}(m, p)$ to $\mathrm{St}(m, p)$ via the map $\pi_{\mathrm{St}(m,p) \mapsto \mathrm{Gr}(m,p)}(\bar{U}^m) = \bar{U}^m \left[ \bar{U}^m \right]^T$. Under this map, an element $d\bar{U}^m \in \mathcal{T}_{\bar{U}^m} \mathrm{St}(m, p)$ horizontal of $\mathcal{T}_{\bar{P}^m} \mathrm{Gr}(m, p)$ must satisfy the condition $d\bar{U}^m = \bar{U}_\perp^m dB$ (we refer readers to Bendokat et al. (2020) for the derivation). Representing gradient as some $dB$ is the most memory efficient method, since the dimension of $\mathrm{Gr}(m, p)$ equals that of $dB$. However, we have to multiply $dB$ with $\bar{U}_\perp^m$ and then with $\bar{D}$ to recover divergence-free velocity bases, while computing either $\bar{U}_\perp^m$ or $\bar{D}$ is intractable. Instead, we choose to work with $d\bar{U}$ directly and rely on the following result that establishes a connection between $dB$ and $d\bar{U}$:

**Lemma C.2.** *For a divergence-free velocity bases $\bar{U}$, a direction $d\bar{U}$ belongs to the tangent plane of $\mathcal{D}(n, p) \cap \mathcal{S}(n, p)$ at $\bar{U}$ if and only if $d\bar{U} \in \mathcal{D}(n, p)$ and $\bar{U}^T d\bar{U} = 0$.*

*Proof.* If $d\bar{U}$ belongs to the tangent plane, then it must satisfy $d\bar{U} = \bar{D} d\bar{U}^m$ for some $d\bar{U}^m = \bar{U}_\perp^m dB$, so $d\bar{U} \in \mathcal{D}(n, p)$. Further, $\bar{U}^T d\bar{U} = \bar{U}^T \bar{D} \bar{U}_\perp^m dB = [\bar{U}^m]^T \bar{U}_\perp^m dB = 0$. Conversely, $d\bar{U} \in \mathcal{D}(n, p)$ implies $d\bar{U} = \bar{D} d\bar{U}^m$ for some $d\bar{U}^m$. Further, $\bar{U}^T d\bar{U} = 0$ implies $\left[ \bar{U}^m \right]^T d\bar{U}^m = 0$, which in turn implies $d\bar{U}^m = \bar{U}_\perp^m dB$ for some $dB$. $\square$

Suppose we have a loss function $\mathcal{L} \circ \bar{v}^+(\bar{v}, \bar{P}^m)$ with $\bar{v}$ as the constant, we can composite the loss function with the map $\pi_{\mathrm{St}(n,p) \mapsto \mathrm{Gr}(m,p)}(\bar{U}) = \bar{D}^T \bar{U} \bar{U}^T \bar{D} = \bar{P}^m$. The domain of this composite function is the intersection of $\mathcal{D}(n, p)$ and $\mathrm{St}(n, p)$, which is an embedded sub-manifold of $\mathbb{R}^{n \times p}$. In order to calculate the gradient on the manifold, we can smoothly extend the composite function to the entire $\mathbb{R}^{n \times p}$, calculate the Euclidean-space gradient denoted by $G \in \mathbb{R}^{n \times p}$, and then project the gradient onto the tangent space. Such projection is defined by Lemma C.2 as:

$$\nabla_{\bar{U}} \mathcal{L} = \bar{P}_\perp \bar{D} \bar{D}^T G, \tag{11}$$

where multiplying by $\bar{D} \bar{D}^T$ ensures $\nabla_{\bar{U}} \mathcal{L} \in \mathcal{D}(n, p)$ and multiplying by $\bar{P}_\perp$ ensures $\bar{U}^T \nabla_{\bar{U}} \mathcal{L} = 0$. Note that, although computing the entire $\bar{D}$ is intractable, evaluating $\bar{D} \bar{D}^T G$ is tractable. Indeed, this involves projecting each column of $G$ into the divergence-free vector subspace, which can be calculated by solving a discrete Poisson's equation Petrila & Trif (2004) via a sparse linear solve at a complexity of $\mathcal{O}(n^\omega)$ Zhang (1998), where $\omega \geq 1$ depends on the numerical linear system solver. Therefore, the entire projection has a cost of $\mathcal{O}(n^\omega p)$, as compared with the complexity of computing $\bar{D}$ being $\mathcal{O}(n^\omega m)$. We refer readers to Appendix C.1 for the derivation of Euclidean space gradient $G$. The computation of $\nabla_{\bar{U}} \mathcal{L}$ over a long trajectory with $T \gg p$ timesteps is rather efficient. Indeed, we can precompute and accumulate $G$ for each timestep, and finally apply divergence-free projection operator to compute $\nabla_{\bar{U}} \mathcal{L}$, the total cost of which is $\mathcal{O}(n^\omega p + Tnp + Tp^3)$.

## C.1 DERIVATIVE FORMULATION IN EUCLIDEAN SPACE

We derive the formula for $G$ in the following lemma:

**Lemma C.3.** *If we introduce the third order tensor:*

$$
\Phi_{\alpha\beta\gamma} \triangleq \sum_{ij} C(e_\beta, \bar{U}_i, \bar{U}_j)\delta_{\alpha\gamma}\frac{z_i^+ + z_i}{2}\frac{z_j^+ + z_j}{2} + \sum_j C(\bar{U}_\alpha, e_\beta, \bar{U}_j)\frac{z_\gamma^+ + z_\gamma}{2}\frac{z_j^+ + z_j}{2} +
$$

$$
\sum_i C(\bar{U}_\alpha, \bar{U}_i, e_\beta)\frac{z_i^+ + z_i}{2}\frac{z_\gamma^+ + z_\gamma}{2},
$$

*and consider an arbitrary differentiable function $\mathcal{L}(\bar{v})$, then the Euclidean space gradient $G$ of function $\mathcal{L} \circ \bar{v}^+(\bar{v}, \bar{U})$ with respect to $\bar{U}$ is defined as:*

$$
G = \bar{v}\nabla\mathcal{L}^T\bar{U}[I + \delta t\nabla\mathbb{C}(z^+)]^{-1}[I - \delta t\nabla\mathbb{C}(z^+)] - \nabla\mathcal{L}^T\bar{U}[I + \delta t\nabla\mathbb{C}(z^+)]^{-1}\Phi +
$$

$$
\nabla\mathcal{L}[z^+ - z]^T - \bar{v}\nabla\mathcal{L}^T\bar{U}. \tag{12}
$$

*Proof.* Assuming $\bar{v}$ is fixed, we first derive some useful fundamental results:

$$
dz = [d\bar{U}^m]^T \bar{D}^T\bar{v} = dB^T[\bar{U}_\perp^m]^T \bar{D}^T\bar{v}
$$

$$
d[\bar{P}\bar{v}] = d[\bar{D}\bar{U}^m z] = \bar{D}d\bar{U}^m z + \bar{D}\bar{U}^m dz =
$$

$$
= \bar{D}[\bar{U}_\perp^m dB[\bar{U}^m]^T + \bar{U}^m dB^T[\bar{U}_\perp^m]^T]\bar{D}^T\bar{v}
$$

$$
= \bar{D}Q^m\left(\begin{matrix} & dB^T \\ dB & \end{matrix}\right)[Q^m]^T\bar{D}^T\bar{v} = d\bar{P}\bar{v} = -d\bar{P}_\perp\bar{v}.
$$

Plugging $\Phi$ into the first-order expansion of Equation 4 and we have:

$$
[I + \delta t\nabla\mathbb{C}(z^+)]dz^+ + \Phi : d\bar{U} = [I - \delta t\nabla\mathbb{C}(z^+)]dz = [I - \delta t\nabla\mathbb{C}(z^+)]d\bar{U}^T\bar{v},
$$

where : denotes tensor contraction of the last two indices. The remaining derivation follows the chain rule:

$$
d\bar{v}^+ = \bar{U}dz^+ + d\bar{U}z^+ + d\bar{P}_\perp\bar{v} = \bar{U}[I + \delta t\nabla\mathbb{C}(z^+)]^{-1}[[I - \delta t\nabla\mathbb{C}(z^+)]d\bar{U}^T\bar{v} - \Phi : d\bar{U}] +
$$

$$
d\bar{U}[z^+ - z] - \bar{U}d\bar{U}^T\bar{v}
$$

$$
d\mathcal{L} = \nabla\mathcal{L}^T d\bar{v}^+ = \mathrm{tr}(d\bar{U}^T G).
$$

By comparing the two sides of the last equation, our result follows. $\square$

## C.2 Alternative Lifted Function

The above derivation is based on the definition of $\bar{v}^+(\bar{v}, \bar{U})$ in Equation 5, which assumes that the orthogonal component of $\bar{v}$ is kept across timesteps. An useful alternative is to assume that the orthogonal component is discarded, which is:

$$
\bar{v}^+(\bar{v}, \bar{U}) \triangleq \bar{U}z^+(\bar{U}^T\bar{v}). \tag{13}
$$

By a similar argument, we can derive the following derivatives for Equation 13:

$$
\frac{\partial\bar{v}^+}{\partial\bar{v}} = \bar{U}[I + \delta t\nabla\mathbb{C}(z^+)]^{-1}[I - \delta t\nabla\mathbb{C}(z^+)]\bar{U}^T
$$

$$
G = \bar{v}\nabla\mathcal{L}^T\bar{U}[I + \delta t\nabla\mathbb{C}(z^+)]^{-1}[I - \delta t\nabla\mathbb{C}(z^+)] - \nabla\mathcal{L}^T\bar{U}[I + \delta t\nabla\mathbb{C}(z^+)]^{-1}\Phi + \nabla\mathcal{L}[z^+]^T.
$$

## D Decoupled Reduced-Order Model

We observe that energy preservation and time-reversibility discussed in Appendix A only requires the tensor $C_{kij}$ to be antisymmetric. In other words, the construction of the tensor $C_{kij}$ via Equation 3 is not necessary. We speculate that using a learned antisymmetric tensor $C_{kij}$ can expose a larger search space, leading to a better match with the full-order model. We denote such model as decoupled reduced-order model, where $C_{kij}$ are separate decision variables not constructed from

| Benchmark | $n$ | $p$ | coupled #variable $(np)$ | decoupled #variable $(np + p^3)$ |
|---|---|---|---|---|
| Taylor Vortices | 8192 | 8/11/16/25 | 65536/90112/131072/204800 | 66048/91443/135168/220425 |
| Plume Rise | 8064 | 6/9/15/26 | 48384/72576/120960/209664 | 48600/73305/124335/227240 |
| Plume Rise+Obstacles | 7416 | 5/8/16/30 | 37080/59328/118656/222480 | 37205/59840/122752/249480 |
| Spherical Plume | 8064 | 36 | 290304 | - |
| Two Plume | 8064 | 59 | 475776 | - |

Table 2: **We summarized the number of decision variables in each example. In the coupled case, our decision parameter is $\bar{U}$ having $np$ variables. In the decoupled case, our decision variables are $\bar{U}, C_{kij}$ having $np + p^3$ variables.**

$\bar{U}$. The formula for $\partial\bar{v}^+/\partial\bar{v}$ Equation 10 stays the same and the formula for $G$ takes the following simpler form:

$$G = \bar{v}\nabla\mathcal{L}^T\bar{U}[I + \delta t\nabla\mathbb{C}(z^+)]^{-1}[I - \delta t\nabla\mathbb{C}(z^+)] + \nabla\mathcal{L}[z^+ - z]^T - \bar{v}\nabla\mathcal{L}^T\bar{U}.$$

Finally, the derivative with respect to $C_{ijk}$ reads:

$$\begin{cases} \mathbb{L} \triangleq [I + \delta t\nabla\mathbb{C}(z^+)]^{-T}\bar{U}^T\nabla\mathcal{L} \\ \frac{\partial\mathcal{L}}{\partial C_{kij}} = \frac{1}{2}\left[\mathbb{L}_k\frac{z_i^+ + z_i}{2}\frac{z_j^+ + z_j}{2} - \mathbb{L}_j\frac{z_i^+ + z_i}{2}\frac{z_k^+ + z_k}{2}\right] \end{cases},$$

where we have projected the derivative onto the antisymmetric subspace. On the downside, there is no universally valid $\delta t$ to make our objective function globally differentiable for all $\bar{U}$ and $C_{kij}$, because discrete time reversibility requires a sufficiently small $\delta t$ that depends on $C_{kij}$. Empirically, however, we have not observed any convergence issue. In Figure 9, we compare the coupled and decoupled versions on the Taylor vortices and the smoke plume benchmark, their convergence histories are almost identical. Therefore, we recommend always using the coupled model due to its theoretical differentiability guarantee. **In Table 2, we summarize the number of decision variables in our various experiments.**

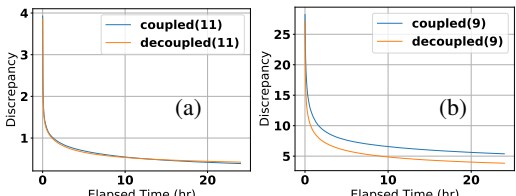

| Benchmark | Loss-Init. | Loss-10k-Iter. |
|---|---|---|
| Taylor Vortices Coupled | 3.93 | 0.39 |
| Taylor Vortices Decoupled | 3.93 | 0.42 |
| Plume Rise Coupled | 28.23 | 5.38 |
| Plume Rise Decoupled | 28.23 | 3.84 |

Figure 9: We compare the performance of coupled and decoupled versions on the Taylor vortices benchmark (a), with $\epsilon = 0.01$ and $p = 11$, and the smoke plume benchmark (b), with $\epsilon = 0.01$ and $p = 9$.

## E  COMPARISON WITH ALTERNATIVE LOSS

**To highlight the effectiveness of our physics correctness loss, we conduct a comparison with two other loss functions: the $\mathcal{L}_1$ and $\mathcal{L}_2$ losses defined as:**

$$\mathcal{L}_1(\bar{v}^+, \hat{v}) \triangleq \|\bar{v}^+ - \hat{v}\|_1 \quad \mathcal{L}_2(\bar{v}^+, \hat{v}) \triangleq \|\bar{v}^+ - \hat{v}\|_2,$$

**where we denote $\hat{v}$ as the velocity generated by the groundtruth fullspace fluid simulator Pavlov et al. (2011). We note that these loss functions are impractical for large-scale test cases because they require solving for the groundtruth data of a different initial condition during each iteration of training. Therefore, we choose to only evaluate them on our first three benchmarks in Table 1, where there is only a single trajectory so $\hat{v}$ can be precomputed. For these benchmarks, we both train and evaluate them on the three losses $\mathcal{L}_{\mathbf{dyn}}, \mathcal{L}_1, \mathcal{L}_2$, and summarize the results in Table 3. We also plot the convergence history of the first benchmark (Taylor Vertices) in Figure 10. Our plots show that, when the first benchmark is trained using $\mathcal{L}_{1,2}$, $\mathcal{L}_{1,2}$ will both decrease by at most $64\%$, but our $\mathcal{L}_{\mathbf{dyn}}$ can increase drastically by at most $1083\%$. Instead, when trained using $\mathcal{L}_{\mathbf{dyn}}$, $\mathcal{L}_{1,2}$ will increase or decrease by at most $3.3\%$ but our $\mathcal{L}_{\mathbf{dyn}}$ can decrease significantly by $76.8\%$. Considering these properties and the fact that $\mathcal{L}_1, \mathcal{L}_2$ is impractical to compute by requiring the groundtruth data, we conclude that our $\mathcal{L}_{\mathbf{dyn}}$ is overall more practical in training reduced fluid systems.**

| Taylor Vertices (p=8) | | | | Plume Rise (p=6) | | | | Plume Rise+Obstacle (p=5) | | | |
|---|---|---|---|---|---|---|---|---|---|---|---|
| Train \ Eval. | $\mathcal{L}_{\text{dyn}}$ | $\mathcal{L}_1(10^{-4})$ | $\mathcal{L}_2(10^{-4})$ | Train \ Eval. | $\mathcal{L}_{\text{dyn}}$ | $\mathcal{L}_1(10^{-4})$ | $\mathcal{L}_2(10^{-4})$ | Train \ Eval. | $\mathcal{L}_{\text{dyn}}$ | $\mathcal{L}_1(10^{-4})$ | $\mathcal{L}_2(10^{-4})$ |
| $\mathcal{L}_{\text{dyn}}$ | 0.58 | 1.23 | 0.27 | $\mathcal{L}_{\text{dyn}}$ | 6.37 | 2.37 | 0.79 | $\mathcal{L}_{\text{dyn}}$ | 10.48 | 1.92 | 0.53 |
| $\mathcal{L}_1$ | 37.81 | 0.44 | 0.04 | $\mathcal{L}_1$ | 57.50 | 0.67 | 0.07 | $\mathcal{L}_1$ | 132.68 | 0.49 | 0.06 |
| $\mathcal{L}_2$ | 165.41 | 0.27 | 0.06 | $\mathcal{L}_2$ | 114.20 | 0.43 | 0.05 | $\mathcal{L}_2$ | 87.83 | 0.69 | 0.06 |

Table 3: **We evaluate our first three benchmarks when trained and evaluated using $\mathcal{L}_{\text{dyn}}, \mathcal{L}_1, \mathcal{L}_2$.**

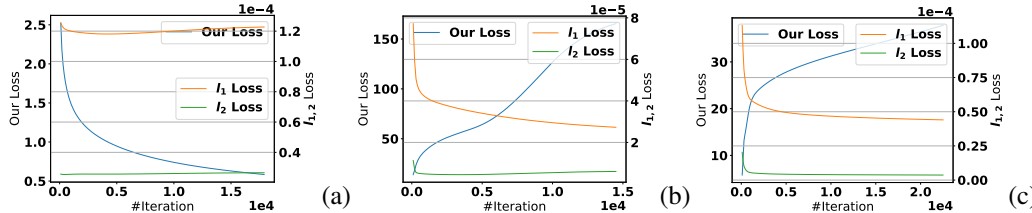

Figure 10: **For our first benchmark (Taylor Vortices), we plot the convergence history when trained using $\mathcal{L}_{\text{dyn}}$ (a), $\mathcal{L}_1$ (b), and $\mathcal{L}_2$ (c). The scale of $\mathcal{L}_{\text{dyn}}$ is shown on the left and $\mathcal{L}_{1,2}$ is shown on the right of each plot.**

## F    COMPARISON WITH DMD

**We have shown that our method works best with POD initialization. In this section, we conduct additional experiments with DMD. DMD extends POD by assuming that the data is generated from a linear dynamic system. DMD can be used both as an intrusive and non-intrusive method. In the intrusive mode, we use DMD to compute a bases $\bar{U}$ and compute $C_{kij}$ from $\bar{U}$ via Equation 3. In the non-intrusive mode, we simply use the DMD-assumed linear dynamic system as the surrogate. To evaluate the performance of DMD, we use two metrics. For the intrusive DMD, we use our physics correctness loss Equation 6. Unfortunately, our physics correctness loss is not suitable for evaluating non-intrusive methods that can be non-reversible. Indeed, it is always possible to let $\mathcal{L}_{\text{dyn}} = 0$ by setting $\bar{v}^+ = \bar{v} = 0$. Therefore, we also measure the energy gain $\Delta e = (\|\bar{v}^T\| - \|\bar{v}^0\|)/\|\bar{v}^0\|$ as an indication of dynamic system stability.**

**We perform the experiments using the open source DMD library Demo et al. (2018) on our first three benchmarks. Their results are shown in Table 4. The results show that the performance of intrusive DMD is worse than either POD or our method, in terms of the physics correctness loss. This is because the main assumption of DMD, i.e., the dynamic system being linear, is invalid for the bilinear dynamic system Equation 3. Instead, POD does not make any assumption on the time dependency between frames and serves as a better initialization for our method. On the other hand, the non-intrusive DMD leads to better performance in terms of $\mathcal{L}_{\text{dyn}}$ but the dynamic system tends to be rather unstable due to a drastic energy gain of $1.9 \times -73.3 \times$.**

| Benchmark | | $\mathcal{L}_{\text{dyn}}/\Delta e$ | | | |
|---|---|---|---|---|---|
| | $p$ | POD | I-DMD | NI-DMD | Ours |
| Taylor Vortices | 8 | 4.84/0 | 20.81/0 | 4.77/4.77 | 0.58/0 |
| Plume Rise | 6 | 57.04/0 | 127.18/0 | 2.21/1.94 | 6.37/0 |
| Plume Rise+Obstacle | 5 | 133.16/0 | 245.20/0 | 2.32/73.3 | 10.48/0 |

Table 4: **We compare the POD baseline and our method with intrusive DMD (I-DMD) and non-intrusive DMD (NI-DMD) in terms of trajectory-wise physics correctness loss and energy gain.**

## G    COMPARISON WITH PINNS

**We conduct comparisons with PINNs Raissi et al. (2019). PINNs was originally designed for solving PDEs, while our divergence-free Navier-Stokes equation is an DAE. In order to extend PINNs to handle DAE, we learn a neural network DAE solution function, denoted as**

$\mathbf{NN}(x, y, t) = (v_x, v_y, \lambda)$ **and represented as an MLP with 3 hidden layers each having** $H$ **neurons and Tanh activation function, and minimize the following physics violation loss:**

$$\|\dot{v} + \nabla \times v \times v + \nabla \lambda\|^2 + \|\nabla \cdot v\|^2.$$

**We also enforce additional temporal and spatial boundary conditions as loss functions. All the loss functions have weights equal to 1. For fairness of comparison, we use the same training data for both our method and PINNs. Note that our method uses grid-based spatial discretization, so we use all the grid centers as spatial samples of training data and we sample the temporal domain at a regular interval of** $\delta t = 0.01$**, which equals to our timestep size. We aim to predict a trajectory of the same length as our method, i.e.** $T\delta t$**. We use Adam as our optimizer and we train both methods on CPU for** $24$ **hours. Since PINNs can lead to non-divergent-free velocity fields, we measure the accuracy of both methods via three metrics:** $\mathcal{L}_{\mathrm{dyn}}$**,** $\Delta e$**, and average divergence error:** $\|\bar{v} - \bar{v}^*\|_\infty$ **where** $\bar{v}^*$ **is the closest divergence-free velocity field to** $\bar{v}$**. The results are summarized in Table 5.**

| Benchmark | PINNs($H = 64$) | | PINNs($H = 128$) | | Ours | |
|---|---|---|---|---|---|---|
| | $\mathcal{L}_{\mathrm{dyn}}/\Delta e$ | $\|v - v^*\|_\infty$ | $\mathcal{L}_{\mathrm{dyn}}/\Delta e$ | $\|v - v^*\|_\infty$ | $\mathcal{L}_{\mathrm{dyn}}/\Delta e$ | $\|v - v^*\|_\infty$ |
| Taylor Vortices | 17.46/0.32 | 0.000027 | 8.96/0.64 | 0.000018 | 0.58/0 | 0 |
| Plume Rise | 6.74/1.06 | 0.005159 | 6.29/0.91 | 0.004466 | 6.37/0 | 0 |

Table 5: **We compare our method and PINNs in terms of** $\mathcal{L}_{\mathrm{dyn}}$ **and** $\|\bar{v} - \bar{v}^*\|_\infty$**.**

**PINNs mostly perform worse than our method in terms of** $\mathcal{L}_{\mathrm{dyn}}$**. In the Taylor Vortices benchmark using** $H = 128$**, the** $\mathcal{L}_{\mathrm{dyn}}$ **metric generated by PINNs is slightly better than our method. But this is again because** $\mathcal{L}_{\mathrm{dyn}}$ **is only designed for measuring time-reversible flows, which is not an effective metric for comparing reversible and non-reversible flows due to its trivial solutions. Such trivial solutions are indeed exhibited in PINNs, as illustrated in Figure 11. After a very short period of time, the solution predicted by PINNs become significantly smeared out and meaningless.**

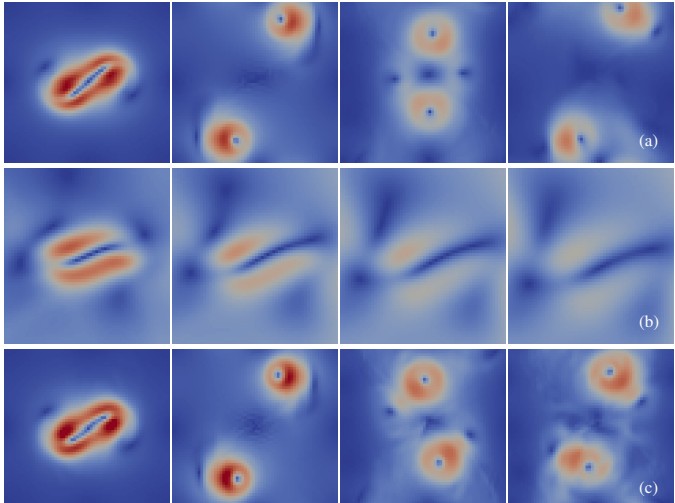

Figure 11: **We compare frames generated by groundtruth (a), PINNs(** $H = 128$ **) (b) and our method (** $\epsilon = 0.0001, p = 25$ **) (c) on the Taylor Vortices benchmark. After very short time period, the results generated by PINNs become significantly smeared out and meaningless.**

## H    ADDITIONAL RESULTS

We demonstrate additional experimental results. Some snapshots of our 4 benchmark scenarios are shown in Figure 12, 13, 14, 15, and 16, respectively.

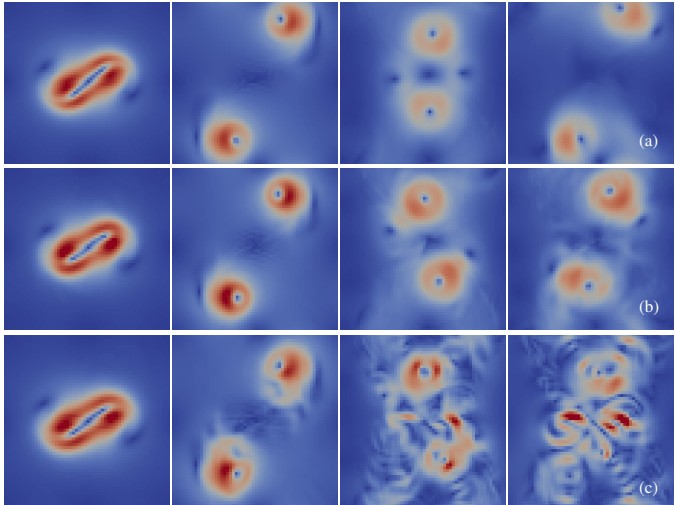

Figure 12: Velocity magnitude field snapshots of the Taylor vortices benchmark, generated by full-order model (a), our method with $\epsilon = 0.0001$ and $p = 25$ (b), and POD with $\epsilon = 0.0001$ and $p = 25$ (c).

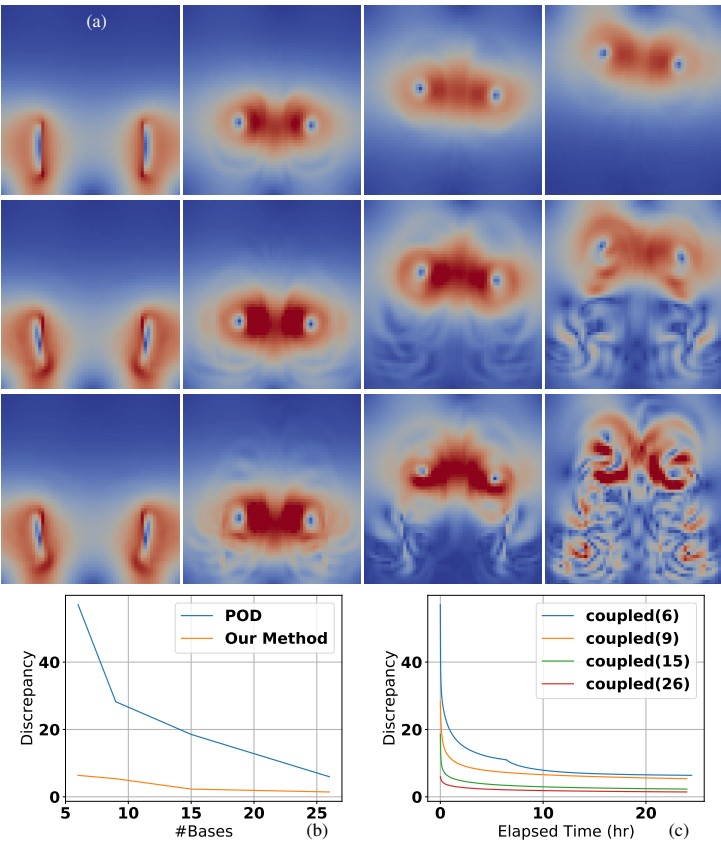

Figure 13: (a) Velocity magnitude field snapshots of the smoke plume benchmark, generated by full-order model (top row), our method with $\epsilon = 0.0001$ and $p = 26$ (middle row), and POD with $\epsilon = 0.0001$ and $p = 26$ (bottom). (b) Trajectory-wise discrepancy loss of POD and our method, under different $p$. (c) The convergence history of our method over 24 hours.

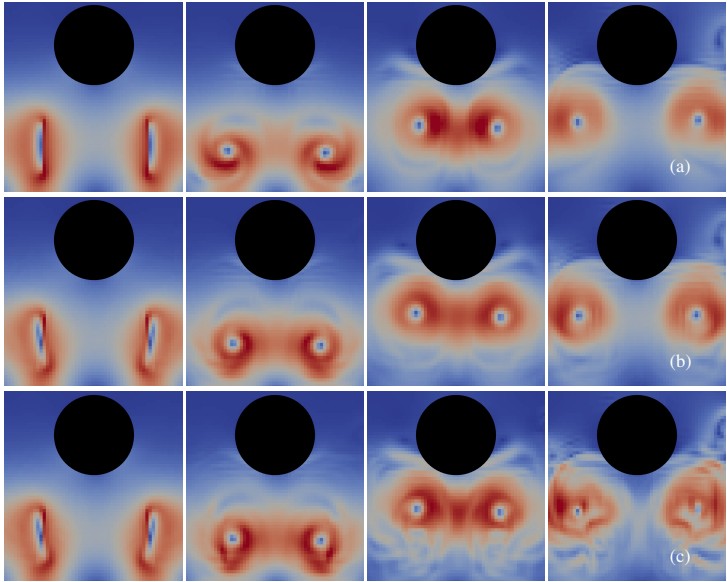

Figure 14: Velocity magnitude field snapshots of the smoke plume benchmark with an spherical obstacle, generated by full-order model (a), our method with $\epsilon = 0.0001$ and $p = 26$ (b), and POD with $\epsilon = 0.0001$ and $p = 26$ (c).

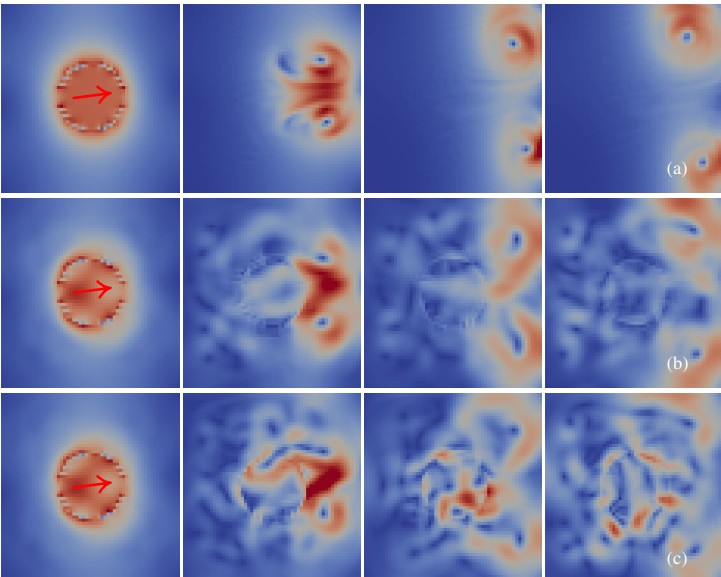

Figure 15: Velocity magnitude field snapshots of the spherical plume benchmark, generated by full-order model (a), our method with $\epsilon = 0.01$ and $p = 36$ (b), and POD with $\epsilon = 0.01$ and $p = 36$ (c). The plume moves along $\theta = 7.5°$ (arrow), which is not covered by our training dataset.

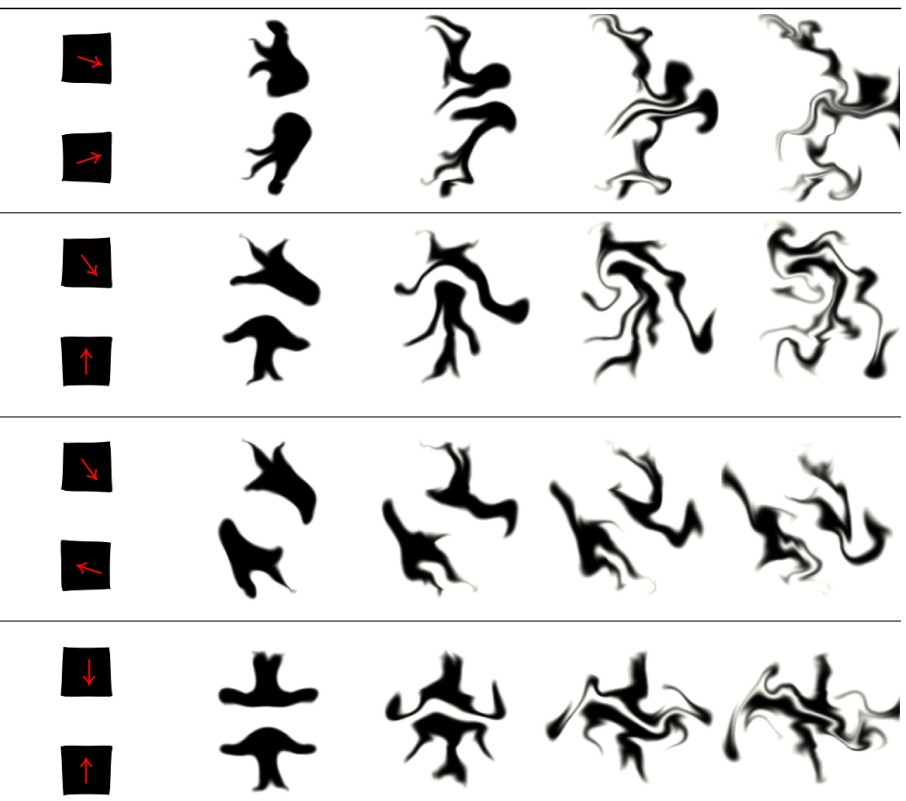

Figure 16: Fluid dye field snapshots of the two cubical plume benchmark generated by our method (time flows to the right). The fluid dye field is initialized as the two cubes (black) and passively advected by the velocity field. With $\epsilon = 0.01$ and only $p = 59$ bases, we can predict a family of trajectories with the two plumes moving at different directions (arrow).

