# OpenReview forum: "Learning Reduced Fluid Dynamics"
_ICLR.cc/2023/Conference — Submitted to ICLR 2023_

### Official Review · Reviewer_Zt39 · 2022-10-24

**Confidence:** 4
**Correctness:** 3
**Technical Novelty And Significance:** 2
**Empirical Novelty And Significance:** Not applicable
**Recommendation:** 3

**Clarity, Quality, Novelty And Reproducibility:**

I think the writing of this paper could be significantly improved. First of all, I found the title of the paper misleading: there is very little "learning" going on. Rather, as explained later on in the text, the paper focus on a data-driven fine tuning of a basis.

Also, the related work is a strange read considering how the proposed algorithm works. Lagrangian approaches are discussed in detail, while the large body of Eulerian approaches for learning fluids is mostly left out. E.g., the divergence-free learning from Tompson et al. 2017, or the learned representations from the DeepFluids Kim et al. 2019 seem much more relevant than graph networks.


**Strength And Weaknesses:**

As strengths of the paper I see the clear reductions in terms of reconstruction errors compared to regular POD approaches. This is clearly demonstrated, and it's neat to see how much difference this fine tuning step can make. Also, I'm not aware of existing works using this  optimization procedure.

As weaknesses, I primarily see the narrow scope of the work, and its somewhat detached nature from learning approaches. The work relies on a Eulerian DEC discretization for reversible flows, instead of targeting general flow-like, divergence free motions. These are not too widely used, as far as I can tell. In addition, the focus on improving POD methods further limits the scope of applications. POD methods are popular as baselines, and for special applications, but not really state of the art. In addition, the clear lack of comparisons with learned baselines highlights that the focus on basis constructions is not a big topic at ICLR. From an ICLR paper I would have expected links and evaluations of how this method fares in the context of other learning-based approaches. This is mentioned in the outlook, but unfortunately missing in the current version.

I would also recommend to better justify the focus on reversible flows. As mentioned above they're not widely used, and not very relevant for game applications. After reading about games as targeted areas of applications in the introduction I was expecting the paper to take a different direction.


**Summary Of The Paper:**

The paper proposes to refine orthogonal bases, as commonly used for "proper orthogonal decomposition" (POD) algorithms with a stochastic, data driven optimization. The paper focuses on a class of reversible, circulation-based flow descriptions. The improvements in terms of reconstruction errors of the optimized basis are evaluated for Taylor vortices and different rising plume variations.


**Summary Of The Review:**

That being said, I think the direction of the paper is interesting. Improved orthogonal bases could definitely find applications, despite potentially being not up-to-par with learned representations. In its current state, I think the work is not really suitable for ICLR, but potentially other more CFD focused venues. For ICLR, I would expect a stronger focus on learning, and actual demonstrations that the proposed method is up to par with the current state of the art for learned fluid simulations.

---

### Official Review · Reviewer_J4j9 · 2022-10-26

**Confidence:** 3
**Correctness:** 3
**Technical Novelty And Significance:** 3
**Empirical Novelty And Significance:** 3
**Recommendation:** 5

**Clarity, Quality, Novelty And Reproducibility:**

The paper introduces quite complex topics, it is somehow hard to understand what is novel and what does already exist. Algorithms help a lot to understand the approach

**Strength And Weaknesses:**

Strengths:
-	Learning reduced fluid dynamics is a very relevant research direction
-	The idea is fundamentally interesting, i.e. to use automatic differentiation to optimize numeric algorithms which have pre-calculated parameters that are used as initialization.
-	I appreciate the time reversible fluid dynamics section, and the transition to reduced model optimization.
-	The results over the baseline look quite convincing, where it is however hard to place the baseline.

Weaknesses:
My general comment is that it is hard to place this work both wrt to existing numerical methods, but also with respect to neural solvers, neural surrogates. That makes it really tough to properly weigh the pros and cons wrt to e.g. Proper Orthogonal Decomposition, Dynamic Model Decomposition, but also operator learning methods or PINNs. Furthermore, I would like to see a test against a model which does not preserve time reversibility.
-	The only comparison is against Proper Orthogonal Decomposition (POD) which as stated in the introduction “is flawed in that it ignores the temporal dependence of state variables”. Naturally the question is for example how Dynamic Model Decomposition (DMD) is doing for the problems.
-	Also from a Deep Learning perspective, e.g. Figure 4 could be learned by an operator learning that takes the initial state as input and outputs the state of the system after e.g. every 100th step. On the other hand, if done with e.g. PINNs this could be performed completely without training data. Just using the boundary conditions for the loss and the equation for the residual loss. It would be really interesting to have a comparison wrt to speed and accuracy.
-	PyTorch and more specifically the automatic differentiation capability of Pytorch is used as an optimization tool. It is however hard to judge from the paper what exactly the parameters are that are optimized. How many parameters are optimized for the different problems? Algorithms 1 and 2 in the appendix help a great deal but it took me a great deal to scroll back and forth. Is it possible to place the algorithms in the main paper and write refer to the important parts in Section 3,4?
-	The loss should be a bit more central to the writing of the paper. In Fig 1 the most important components are the reduced-order fluid model and the trajectory-wise discrepancy loss. However, Sections 4.3 takes a huge part of the main paper and is actually hard to follow. In my opinion the readability of Section 4 can be improved by focusing more on the relevant parts and not let the reader figure out what they are.
-	Why is e.g. JAX not used for optimization? Did the authors consider that?
-	How do optimization time vs inference time relate between the different models?
-	How do the individual components of the algorithm relate to the performance, e.g. how does the performance change for other loss terms?
-	Is it correct that for the first benchmark a single trajectory is used, whereas in the second benchmark more trajectories are used for optimization? On which trajectories is the testing done afterwards? Shouldn’t there be more than just one trajectories to get a better performance estimate?



**Summary Of The Paper:**

The paper introduces a machine learning approach to identify model-reduced dynamical systems. The main idea is to use stochastic (Riemann) optimization which minimizes the expected trajectory-wise reduction error over a distribution of initial conditions. This is achieved by formulating the fluid dynamics problem as an invertible state transfer function.
The paper identifies fluid velocities as a point on a manifold where the reduced trajectory depends on the choice of the subspace of fluid velocities. This map from subspaces to reduced trajectories is shown to be globally differentiable.
The paper compares against Proper Orthogonal Decomposition as an example of data-driven sub-manifold model.



**Summary Of The Review:**

The idea and the topic in general are quite relevant.  For the current version of the paper, it is however hard to place this work both wrt to existing numerical methods, but also with respect to neural solvers, or neural surrogates. That makes it really tough to properly weigh the pros and cons wrt to e.g. Proper Orthogonal Decomposition, Dynamic Model Decomposition, but also operator learning methods or PINNs.

---

### Official Review · Reviewer_eGGS · 2022-10-31

**Confidence:** 3
**Correctness:** 4
**Technical Novelty And Significance:** 3
**Empirical Novelty And Significance:** 3
**Recommendation:** 8

**Clarity, Quality, Novelty And Reproducibility:**

*Quality:* The submission appears to be technically sound, and the authors seem forthcoming about the limitations of the work.

*Clarity:* The submission is clearly written and well organized.

*Originality:* The work adapts an existing approach for learning a reduced order model to the nonlinear setting, which appears nontrivial, though I am not certain of this. The work is well situated in the existing literature.

**Strength And Weaknesses:**

The authors address the challenging problem of predicting the state evolution of high-dimensional, time-reversible, nonlinear fluid dynamic systems; however, I am uncertain how to understand the magnitude of the contribution. While they demonstrate strong performance against POD; as I understand, they also initialize with POD. I am also unsure why the authors choose to evaluate against a nondescript POD baseline rather than DEIM or another intrusive reduced order method noted in the related work. Moreover, as the authors mention, the method takes considerably longer to run than the baseline approach.

**Summary Of The Paper:**

The authors propose a method to learn the time-reversible, nonlinear dynamics of an incompressible fluid. For computational tractability, the authors learn a reduced model over a limited set of initial conditions with tensor precomputation. The authors show that their approach predicts solutions significantly closer to the full order method than conventional methods.

**Summary Of The Review:**

The paper is clearly written and well organized, and appears to extend an existing method in a nontrivial way.

---

### Decision · Program_Chairs · 2023-01-20

**Decision:**

Reject

**Justification For Why Not Higher Score:**

The method should be better positioned in the literature, and clarify if it allows for reasonable interpolation and generalization.

**Justification For Why Not Lower Score:**

The idea is in its principle quite interesting, all reviewers agreed on that.

**Metareview: Summary, Strengths And Weaknesses:**

The paper proposes a method for the identification of model-reduced dynamical systems, minimizing errors of trajectories over a distribution of initial conditions. To achieve this, the paper formulates fluid dynamics modelling as an invertible state transfer function problem. The paper raised some external and internal discussion. The questions, after the rebutal where

- are the results state-of-the-art, and
- is the method relevant to the community?
- is the paper overclaiming?

Regarding question #2, it is not clear if the proposed method can interpolate and generalize, posing a clear limitation  for the method since like POD, it is a compression scheme that can play back a dataset. While initially there was a question if it is a problem that no NNs are used, in my personal opinion that is not really a problem for as long as there are learned compoments; after all, there is backpropagation. However, it is a problem that scope is really only the refined construction of the basis and transition tensor. So far, given the progress in learned surrogates (not so much PINNs, but more Conv-Net-based time stepping) that also interpolate and generalize, it is important to position the method and the results into the context of existing methods better and more convincingly.